# Caregiver acceptability of the guidelines for managing young infants with possible serious bacterial infections (PSBI) in primary care facilities in rural Bangladesh

Jennifer A. Applegate[1], Salahuddin Ahmed[2], Meagan Harrison[1], Jennifer Callaghan-Koru[3], Mahfuza Mousumi[4], Nazma Begum[2], Mamun Ibne Moin[2], Taufique Joarder[1], Sabbir Ahmed[5], Joby George[5], Dipak K. Mitra[6], ASM Nawshad Uddin Ahmed[7], Mohammod Shahidullah[8], Abdullah H. Baqui[1]*

1 Department of International Health, Bloomberg School of Public Health, Johns Hopkins University, Baltimore, Maryland, United States of America, 2 Johns Hopkins University-Bangladesh, Dhaka, Bangladesh, 3 Department of Sociology, Anthropology, and Health Administration and Policy, University of Maryland, Baltimore County, Baltimore, Maryland, United States of America, 4 Jhpiego Bangladesh, Dhaka, Bangladesh, 5 USAID's MaMoni Health Systems Strengthening Project, Save the Children, Washington, DC, United States of America, 6 Department of Public Health, School of Health and Life Sciences, North South University, Dhaka, Bangladesh, 7 Department of Pediatrics, Child Health Research Foundation (CHRF), Dhaka, Bangladesh, 8 Department of Neonatology, Bangabandhu Sheikh Mujib Medical University (BSMMU), Dhaka, Bangladesh

* abaqui@jhu.edu

## Abstract

### Introduction

Many infants with possible serious bacterial infections (PSBI) do not receive inpatient treatment because hospital care may not be affordable, accessible, or acceptable for families. In 2015, WHO issued guidelines for managing PSBI in young infants (0–59 days) with simpler antibiotic regimens when hospital care is not feasible. Bangladesh adopted WHO's guidelines for implementation in outpatient primary health centers. We report results of an implementation research study that assessed caregiver acceptability of the guidelines in three rural sub-districts of Bangladesh during early implementation (October 2015-August 2016).

### Methods

We included 19 outpatient primary health centers involved in the initial rollout of the infection management guidelines. We extracted data for all PSBI cases (N = 192) from facility registers to identify gaps in referral feasibility, simplified antibiotic treatment, and follow-up. Focus group discussions (FGD) and in-depth interviews (IDI) were conducted with both caregivers (6 FGDs; 23 IDIs) and providers (2 FGDs; 28 IDIs) to assess caregiver acceptability of the guidelines.

### Results

Referral to the hospital was not feasible for many families (83.3%; N = 160/192) and acceptance varied by infection severity. Barriers to referral feasibility included economic and

**Data Availability Statement:** Data are available from ICPSR through the University of Michigan at the following DOI: https://doi.org/10.3886/E118382V1.

**Funding:** Funding information- AB received support for this study from the United States Agency for International Development (USAID) through JHU's Health Research Challenge for Impact (grant number GHS-A00-0900004-00). The contents are the responsibilities of the authors and do not necessarily reflect the views of USAID, the United States Government and/or the decisions, policy, or views of their respective organizations.

**Competing interests:** NO authors have competing interests.

household factors, and previous experiences with poor quality of care at the sub-district hospital. Conversely, providers and caregivers indicated high acceptability of simplified antibiotic treatment. 80% (N = 96/120) of infants with clinical severe infection for whom referral was not feasible returned to the facility for the second antibiotic injection. Some providers reported developing local solutions—including engaging informal providers in treatment of the infant—to address organizational barriers and promote treatment compliance. Follow-up of young infants receiving simplified treatment is critical, but only 67.4% (N = 87/129) of infants received fourth day follow-up. Some providers' reported deviations from the guidelines that shifted responsibility of follow-up to the caregiver, which may have contributed to lapses.

## Conclusion

Caregivers' perception of trust and communication with providers were influential in caregiver acceptability of care. Few caregivers accepted referral to the sub-district hospital, suggesting low acceptability of this option. When referral was not feasible, many caregivers reported satisfaction with simplified antibiotic treatment. Local solutions described by providers require further examination in this context to assess the safety and potential value of these strategies in outpatient treatment. Our findings suggest strengthening providers' interpersonal skills could improve caregiver acceptability of the guidelines.

## Introduction

Bangladesh has made tremendous progress in reducing sepsis-related deaths during the neonatal period. Between 1990 and 2015, newborn deaths due to sepsis reduced by more than half, which is largely attributable to interventions to prevent community-acquired newborn infections in this context, including promotion of clean delivery, essential newborn care practices, and improved access to timely treatment with antibiotics [1–6]. Appropriate case management of serious infections in newborns, including early identification and appropriate treatment with antibiotics, may result in a 24% reduction in infection-related neonatal deaths [3, 7–9]. However, implementing these interventions at scale is challenging, resulting in suboptimal coverage [10]. As a result, newborn infections—including sepsis, meningitis, and pneumonia—remain a major contributor to neonatal morbidity and mortality in Bangladesh and globally [6, 11, 12].

The World Health Organization (WHO) recommends that all young infants (0–59 days) with possible serious bacterial infections (PSBI) be referred to hospitals and treated with a 7 to 10-day course of a combination of gentamicin and either penicillin or ampicillin [13]. In resource-limited settings, however, many infants with PSBI may not receive the recommended in-patient care due to limited access to hospital facilities, social and cultural practices that may impede care-seeking, financial constraints, and low acceptability of hospital care [13–16]. The revised guidelines provide guidance on outpatient management of PSBI in young infants with simplified antibiotic regimens—including fewer injections combined with oral antibiotics—at primary healthcare facilities closer to the community [13].

Bangladesh was one of the first countries to adopt the WHO recommendations [17]. In 2015, the government of Bangladesh partnered with funding agencies, implementation groups, and research organizations to operationalize the guidelines in primary health facilities in a few

selected districts [17]. An implementation research study was embedded in program rollout to document lessons and inform nationwide scale-up. The updated infection management guidelines aim to increase coverage of treatment for newborn infections through provision of public sector care that are more affordable, accessible, and acceptable for families [13, 14, 18].

In Bangladesh, healthcare services are sought from a mix of sources including public and private providers in both the formal and informal sector, as well as traditional medicine [19–21]. Rural and disadvantaged populations in Bangladesh commonly first seek care from a wide array of informal providers including village doctors, traditional healers, drug sellers, and homeopathic doctors [18, 19, 22, 23]. Village doctors are prominent providers in rural communities—approximately 12.5 per 10,000 population. Most village doctors have limited or no standard training and are connected to unlicensed pharmacies where they diagnose patients and sell prescription medicines [18, 19]. Public healthcare is highly subsidized by the government, requiring minimal or no payments from clients, especially in the outpatient centers targeted for this intervention (i.e., Union Health & Family Welfare Centers [UH&FWC]; catchment area ~25,000 persons) [24, 25]. Despite adequate geographic distribution of these outpatient health centers, UH&FWCs have been largely under-utilized by the communities, and many were not fully functional due to staff shortages, insufficient equipment, poor infrastructure, unavailability of water and electricity, and perceived low quality of care [19, 21, 26–28]. Families often prefer to seek care from the private sector, informal and formal, due to convenience and acceptability of services, but informal providers are highly unregulated and formal care comes at high out-of-pocket costs for families [18, 19, 24, 29]. With the rollout of the new guidelines, there is an urgent need to strengthen existing public sector facilities to improve the quality of care and acceptability of health services for families of sick young infants.

The guidelines are not designed to replace hospital care, which remains the first-line treatment for PSBI and only treatment option for critically ill infants [13]. Bangladesh's national referral system is designed so that patients move up through the pyramid of care, typically beginning at the community level—at the base of the pyramid—with referrals to a larger facility for more specialized care at the sub-district or district-level (Fig 1). Referral is necessary when a patient's condition requires management that supersedes the training and/or resources at the basic level of care [30]. The optimal situation is for patients to receive appropriate care at the lowest level possible to conserve resources for the patient and health system [30, 31]. However, complexities around referral feasibility are well-documented in the literature and barriers may vary by context [30, 32, 33]. The infection management guidelines aim to improve accessibility and quality of care at lower levels of the pyramid when referral is not feasible. WHO also emphasizes the need to strengthen referral linkages, improve the quality of care at referral facilities, and understand context-specific barriers to referral feasibility [13, 14].

Previous analyses from our implementation research study provide insights about quality of care from the supply-side, including the structure of healthcare provision in the UH&FWCs (e.g., drug supply, infrastructure, number and training of providers) and providers' performance on the guidelines over time. In this paper, we present mixed methods findings to assess caregiver acceptability of the infection management guidelines during early implementation (October 2015-August 2016). *Acceptability* as an implementation research outcome differs from the larger construct of *satisfaction* in that it focuses on a particular intervention or treatment—rather than general service experience—and is typically measured based on different aspects of the intervention and through the perspective of various stakeholders [34]. In the early stages of implementation, acceptability is an important precursor to adoption of an intervention and will likely affect penetration and sustainability in the later stages [34, 35]. Previous studies have shown multiple measures are beneficial for assessing client acceptability of health interventions, understanding the interpersonal relationship between provider and client, and

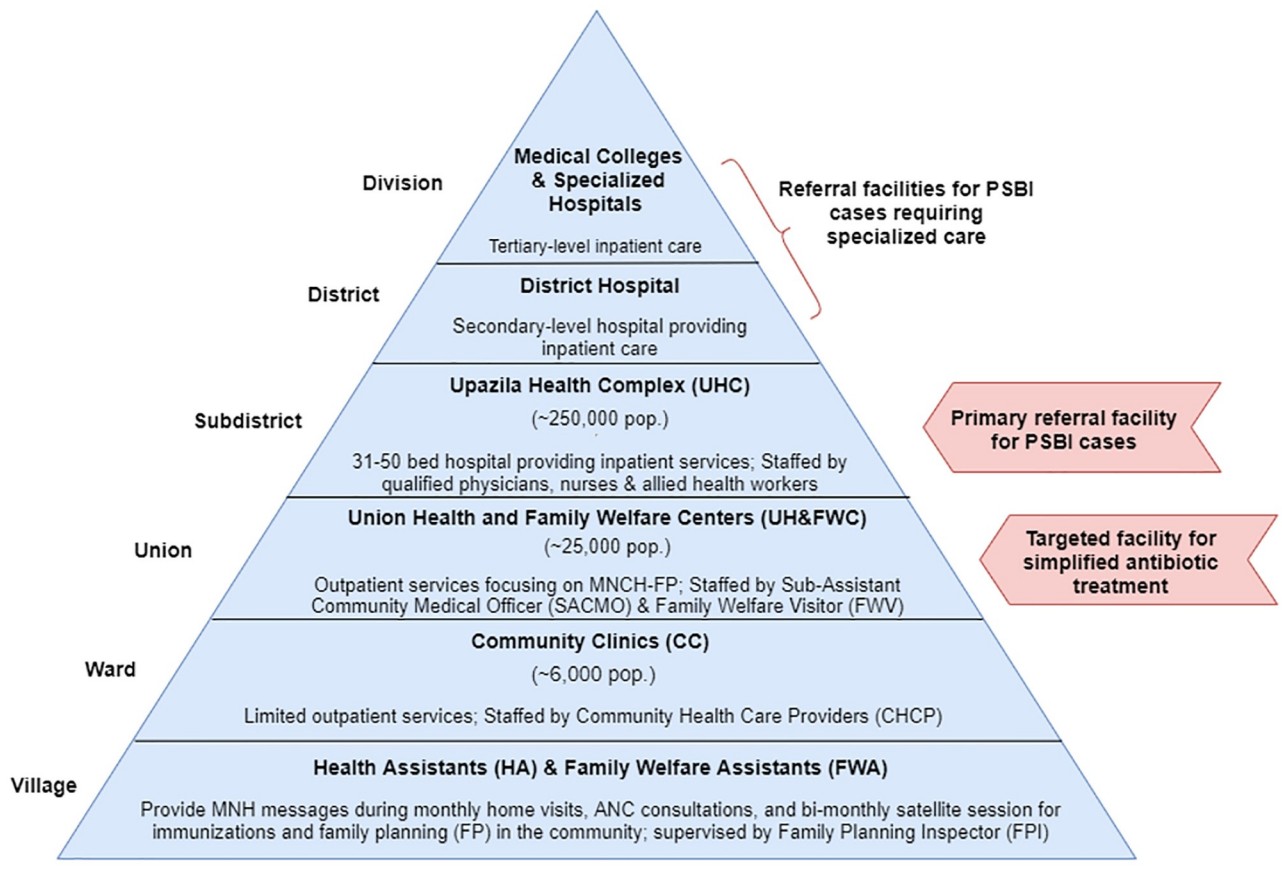

**Fig 1. Provision of public health services relevant to management of PSBI in young infants.**

developing strategies to improve quality of care [27, 34, 36]. We posit that multi-level factors will affect caregiver acceptability of the updated treatment guidelines. Thus, we explored caregiver acceptability through application of the socioecological model (SEM), which accounts for multiple levels of influence on behavior [37]. Here, we discuss caregiver acceptability of the infection management guidelines through identifying gaps in care of PSBI cases and examine barriers and facilitators to acceptability from the perspective of providers and caregivers in three sub-districts of rural Bangladesh.

## Methods

### Context and intervention

Our study area included primary health centers in two sub-districts of Sylhet district in Sylhet division and one sub-district in Lakshmipur district in Chittagong division. Sylhet and Chittagong are historically low performing divisions of Bangladesh for maternal, newborn and child health indicators, including low rates of facility delivery and skilled attendants at birth [38]. Bangladesh's Ministry of Health and Family Welfare (MOHFW) maintains a three-tier system for delivering public healthcare services at all administrative levels and follows the Integrated Management Childhood Illness (IMCI) protocol for management of sick children in primary health facilities [22, 39]. Implementation of the infection management guidelines targeted primary health facilities (i.e., UH&FWCs), which are generally staffed by 2–3 formally trained

providers—the Sub-Assistant Community Medical Officer (SACMO) and the Family Welfare Visitor (FWV) (Fig 1). While some of these facilities have a position for a doctor available, these posts are often vacant [19, 21, 28]. The SACMO has 3 years training on general health-care, including child health, from a Medical Assistant Training School [19]. The FWV has at least 18 months training from a private or government facility on midwifery and contraceptive management [19, 21, 22].

The SACMO is the designated provider for assessing, classifying, and treating young infants according to the infection management guidelines. The guidelines include a clinical algorithm for classifying signs of infection in young infants and provide guidance on antibiotic treat-ment, referral advice and follow-up of sick young infants [25]. Infants with signs of PSBI (fever, hypothermia, convulsions, respiratory rate ≥60 breaths per minute if infant is <7 days, severe chest in-drawing, no movement or movement only upon stimulation, feeding poorly or not feeding at all) are classified as Critical Illness (CI), Clinical Severe Infection (CSI), or Iso-lated Fast Breathing <7 days (IFB <7). Accordingly, the SACMO provides the first dose of antibiotics based on the infant's weight and refers the infants with signs of PSBI (i.e., CI, CSI, and IFB [<7D]) to the sub-district hospital (Upazila Health Complex [UHC]; catchment area ~250,000 persons) for inpatient care [22, 25] (Fig 1). Young infants classified as CSI or IFB (<7D) whose families decline hospital referral are eligible for simplified antibiotic treatment with injectable gentamicin once daily for two days and oral amoxicillin twice daily for seven days. Hospital referral for inpatient treatment is the only option for critically ill infants. Care-givers of infants with CSI that decline referral are instructed to return to the UH&FWC the next day for the 2nd gentamicin injection. The FWV may provide the 2nd injection if the SACMO is not available. The SACMO follows-up with caregivers over telephone on the fourth day, and if the infant's condition has not improved, advises the caregivers to seek care at the UHC. On the eighth day of treatment, the family receives a home visit from the Family Plan-ning Inspector (FPI), who are trained as supervisors of frontline workers (i.e., Family Welfare Assistants), to assess treatment compliance and the condition of the infant [25]. Henceforth, we will refer to SACMOs as "providers," UH&FWCs as "health centers," and the UHC as the "sub-district hospital."

## Design and data collection

Project partners—Projahnmo and MaMoni Health Systems Strengthening (HSS)—supported the MOHFW to implement the guidelines in 3 sub-districts of the selected districts, Sylhet and Lakshmipur respectively. Both implementation partners have extensive experience supporting maternal, newborn and child health interventions and health system strengthening in rural Bangladesh, which has been described previously [40, 41]. The supportive inputs from these two partners primarily focused on improving the performance and capacity of health services, including collaborating with the MOHFW to ensure the implementation readiness of health centers to manage infections in young infants, and supporting training and supervision of pro-viders. Additional details on our implementation strategies and study protocol are available elsewhere [42].

Our study employed a mixed methods approach to data collection following a convergent parallel design where quantitative and qualitative data were collected concurrently [43]. We adapted the SEM to inform the development of interview guides to examine caregivers' accept-ability of key components of the intervention—decision to seek care and perception of public sector care, referral feasibility, simplified antibiotic treatment, and follow-up (Fig 2 & S1 Table). Specifically, three data collection activities were included: 1) weekly extraction of data from facility registers to monitor adherence to the guidelines for referral feasibility, simplified

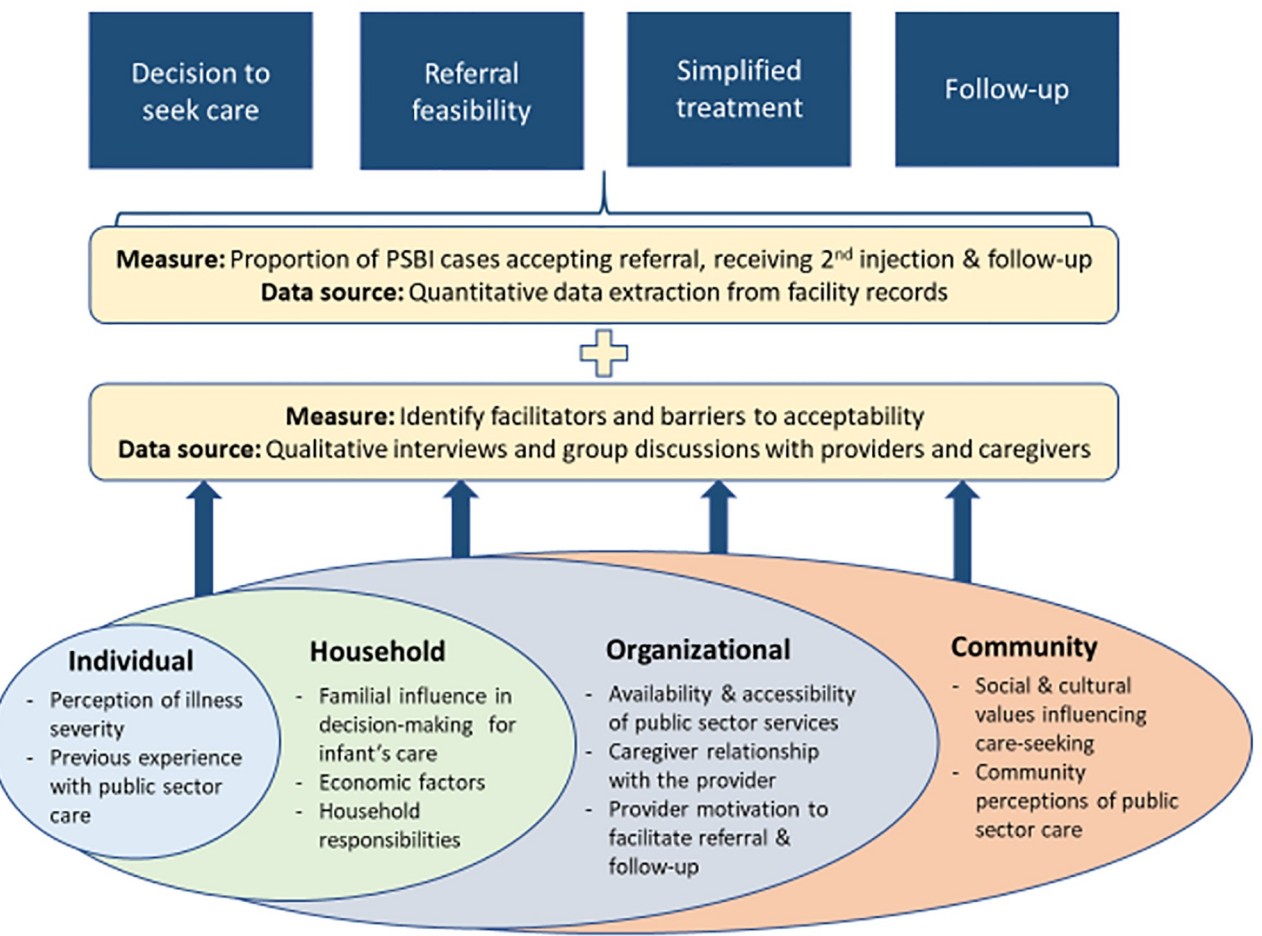

**Fig 2. Adapted socioecological model for analysis of embedded mixed methods study to assess caregiver acceptability of key components of infection management guidelines.**

antibiotic treatment, and follow-up; 2) focus group discussions (FGD) and in-depth interviews (IDI) with facility providers; and 3) FGDs and IDIs with caregivers of infants to examine facilitators and barriers to caregiver acceptability of the infection management guidelines.

Weekly Data Extraction: Data collectors visited the 19 UH&FWCs weekly to abstract data from facility records of all young infants that sought services from October 2015-August 2016. The young infant registers used by providers were developed specifically for the infection management guidelines and distributed to facilities as part of program rollout [25]. Our team adapted the register into an electronic form and recorded data weekly using tablets. For this analysis we included data on the infant's age, provider classification of illness, referral feasibility, antibiotic treatment, and follow-up.

FGDs and IDIs with providers: FGDs and IDIs with providers were conducted in both study areas to assess their perceptions of acceptability of the guidelines using semi-structured interview guides. We piloted the interview guides prior to rollout of the guidelines and adapted these to improve provider comprehension of questions (S1 and S2 Files). The goal of these interviews was to assess the facilitators and barriers to implementation of the guidelines in the health centers. FGDs were conducted at the sub-district hospital on a date that coincided with their monthly meetings or routine collection of medicines from this location. IDIs were

conducted in the health center every 3–4 months during the study period. All providers (N = 19) trained in the guidelines and providing care in the study area health centers were eligible to participate in the interviews. In the final months of data collection, follow-up interviews were conducted with providers to explore themes identified by the study team through the analysis of interviews conducted during early implementation.

FGDs and IDIs with Caregivers: FGDs with caregivers aimed to explore motivators and barriers to care-seeking from primary health facilities including specific questions about their previous experiences with care. The interview guides were piloted by the study team prior to rollout of the guidelines and adapted to improve caregiver comprehension of questions (S4 and S5 Files). For group discussions, caregivers were selected through convenience sampling of mothers (13–49 years) of infants under six months of age who were willing and able to share their care-seeking experiences for their infant's illness. The number of participants for each focus group ranged from 6 to 8 mothers. When selecting caregivers for participation in the FGDs we originally tried to select caregivers of young infants (0–59 days) since this age group is the focus of the intervention. However, we had difficulty recruiting mothers with infants in this age group because many women in rural Bangladesh predominantly spend their time at home the first two months postpartum, which limits their ability to join a group discussion in the community [23, 44]. We adjusted our inclusion criteria to allow mothers of infant under 6 months of age to participate in the FGDs. The goal of our FGDs was to obtain community perceptions about young infant illness, patterns of care-seeking, and perceptions of care provided in government facilities.

The study team used facility records to identify young infants for follow-up in the community and select caregivers for IDIs. Caregivers were purposively selected based on their infant's categorization of infection. We conducted in-depth interviews with caregivers of infants for each category of PSBI. However, we prioritized reaching a point of saturation for the clinical severe infection cases because these infants received referral to the hospital and were eligible for simplified antibiotic treatment, including two gentamicin injections, when referral was not feasible for families. The goal of these interviews was to assess the facilitators and barriers to referral feasibility and simplified antibiotic treatment from the perception of caregivers of young infants receiving treatment according to the guidelines.

All qualitative data collectors were Bangladeshi and conducted the FGDs and IDIs in the local language. Interviews were recorded and transcribed into English by trained translators for analysis. Following the interviews, research assistants also participated in debriefing sessions led by the research officers utilizing a thematic framework. Notes from these debriefing sessions were also translated into English and included in the analysis.

### Ethics statement

Ethical approval was obtained for this study from the Johns Hopkins Bloomberg School of Public Health Institutional Review Board (JHSPH IRB) and the Bangladesh Institute of Child Health (BICH) Review Board. Written informed consent was obtained from all providers in the study, while oral informed consent was obtained from caregivers. Verbal consent was chosen for caregivers due to low literacy rates in this population. To ensure the caregiver understood the study procedures, our data collectors read the consent forms aloud and provided the caregiver an opportunity to ask questions. Prior to the initiation of the interviews, all consenting participants either signed or provided their thumbprint as proof of consent. A witness and member of the research team obtaining verbal consent also signed the consent form. JHSPH IRB and BICH reviewed and approved all consent procedures for this study.

### Analysis

**Quantitative.** Quantitative data were analyzed using Stata version 14 (StataCorp LP). Records were excluded if date of assessment or illness classification were missing. Descriptive results are summarized as frequencies and proportions for referral feasibility, caregiver return to the health center for the second injection, and follow-up according to the guidelines. We estimated the percent of PSBI cases captured at service delivery points in our study area based on the expected annual number of births for both Sylhet and Chittagong Divisions [38] and incidence of PSBI in young infants (95.4/1000) [45] in this setting.

**Qualitative.** We employed an iterative approach to development of the coding framework [46]. The framework was developed using *a priori* codes derived from the interview guides and the research questions related to acceptability of the guidelines and application of SEM. Emergent codes were added to the codebook as necessary to capture themes that were suggested in the data but not initially anticipated in the *a priori* codes. We coded transcripts using the computer software program Dedoose [47]. This study employed analytical methods of continual analysis, coding, and memoing. Our team reviewed transcripts of respondents throughout the study period based on both inductive and deductive themes. We adapted the questionnaire to explore emergent themes [48]. Ultimately, we developed a coding framework—including *a priori* and emergent codes—based on continual review of the qualitative data, which was used for the final analysis. Each transcript was coded using this scheme and charting of the coded passages was used to facilitate interpretation of the data between two researchers.

## Results

### Quantitative

We analyzed data on the infant's age and sex, infection classification, referral decision, and antibiotic treatment for 1052 facility records. Records were excluded if date of assessment (N = 2) or illness classification (N = 99) were missing. We included records of young infants classified with PSBI (i.e., CI, CSI, IFB <7D) that required referral according to the guidelines (N = 192) (Table 1).

Based on expected PSBI incidence for young infants in this context [45], we estimated that only 16.3% [95% CI: 14.4, 18.5] of the expected cases sought care from the study area health centers from October 2015-August 2016. Referral to the hospital was not feasible for many families (83.3%; N = 160/192) and acceptance differed by infection severity (CI [28%; N = 12/43], CSI [14%; N = 20/140], IFB <7D [0%; N = 0/9]) (Fig 3). For infants classified with clinical severe infection receiving simplified antibiotic treatment, 80% (N = 96/120) of caregivers returned to the facility for the second injection. When referral was not feasible for families, 68% (N = 82/120) of infants with CSI and 56% (N = 5/9) of very young infants with IFB received follow-up from the provider on the fourth day of treatment. Our quantitative results indicate that day 8 follow-up in the community by the FPI was low for both CSI (36%; N = 43/140) and IFB (56%; N = 5/9). No caregiver in our interviews reported receiving a day 8 follow-up visit from the FPI for their infant's illness episode, so we were unable to assess caregiver acceptability of these visits.

### Qualitative

We analyzed qualitative data from 6 FGDs, 3 in each study area, and 23 IDIs with caregivers to explore their perceptions of care provided to sick young infants at public primary health facilities (e.g., UH&FWC and UHC) and acceptability of the infection management guidelines. All mothers completing an interview were women of reproductive age (13–49 years old) and

**Table 1. Descriptive characteristics of sick young infants classified as PSBI at health centers.**

| Characteristic | % (n) |
|---|---|
| | **N = 192** |
| **Age (in days)** | |
| <7 days | 27.6% (53) |
| 7–28 days | 28.7% (55) |
| 29–59 days | 43.7% (84) |
| **Sex of infant** | |
| Male | 57.8% (111) |
| Female | 42.2% (81) |
| **Signs of illness recorded by SACMO** | |
| Respiratory rate ≥60/min | 52.1% (100) |
| Severe chest in-drawing | 45.8% (88) |
| Not feeding well | 42.2% (81) |
| Fever (>37.5C) | 27.6% (53) |
| Less movement than normal | 19.3% (37) |
| Unable to feed | 12.5% (24) |
| Unconscious/Drowsy | 9.9% (19) |
| Hypothermia (<35.5C) | 6.3% (12) |
| Convulsions or history of convulsions | 4.2% (8) |
| Persistent Vomiting | 3.6% (7) |
| Umbilicus redness | 2.1% (4) |
| Weight <1500 g | 2.1% (4) |
| Bulging fontanelle | 1.6% (3) |
| Central cyanosis | 1.6% (3) |
| Other signs | 0.5% (1) |
| Skin pustules | 0% (0) |

received care for their young infant according to the infection management guidelines. To understand providers' perspective of care provision, we conducted two FGDs with providers in the early months of the study (November and December 2015), 19 interviews during the study period, and nine follow-up interviews in the final months of the study. All providers in the study area participated in at least one interview during the study period. Our qualitative findings are presented in four sub-sections: 1) decision to seek care and experience at public health centers; 2) referral feasibility for families of infants with PSBI; 3) simplified antibiotic regimen and caregiver return for second day injection; 4) follow-up on the fourth day for infants receiving outpatient treatment. Table 2 presents our mixed methods findings around acceptability for each component included in this analysis.

**Decision to seek care and experience at public health centers.** According to caregivers and providers, mothers' autonomy to seek care for their infant outside the home is limited; oftentimes she must first obtain the consent of her husband. As one mother noted, "I won't be able to go outside of home without my husband's permission." Rather, the decision about when and where to seek care for the infant was described as a collective process that frequently included the baby's mother, father, and/or grandparents (maternal or paternal). Informal providers residing in the community, especially village doctors, were often cited by caregivers in group discussions and interviews as the first source of care because services are provided at a reduced cost (e.g., no consultation fee, shorter distance to travel, no wait times) and they have a good relationship with the family. Of the village doctor, one mother said, "He calls us by

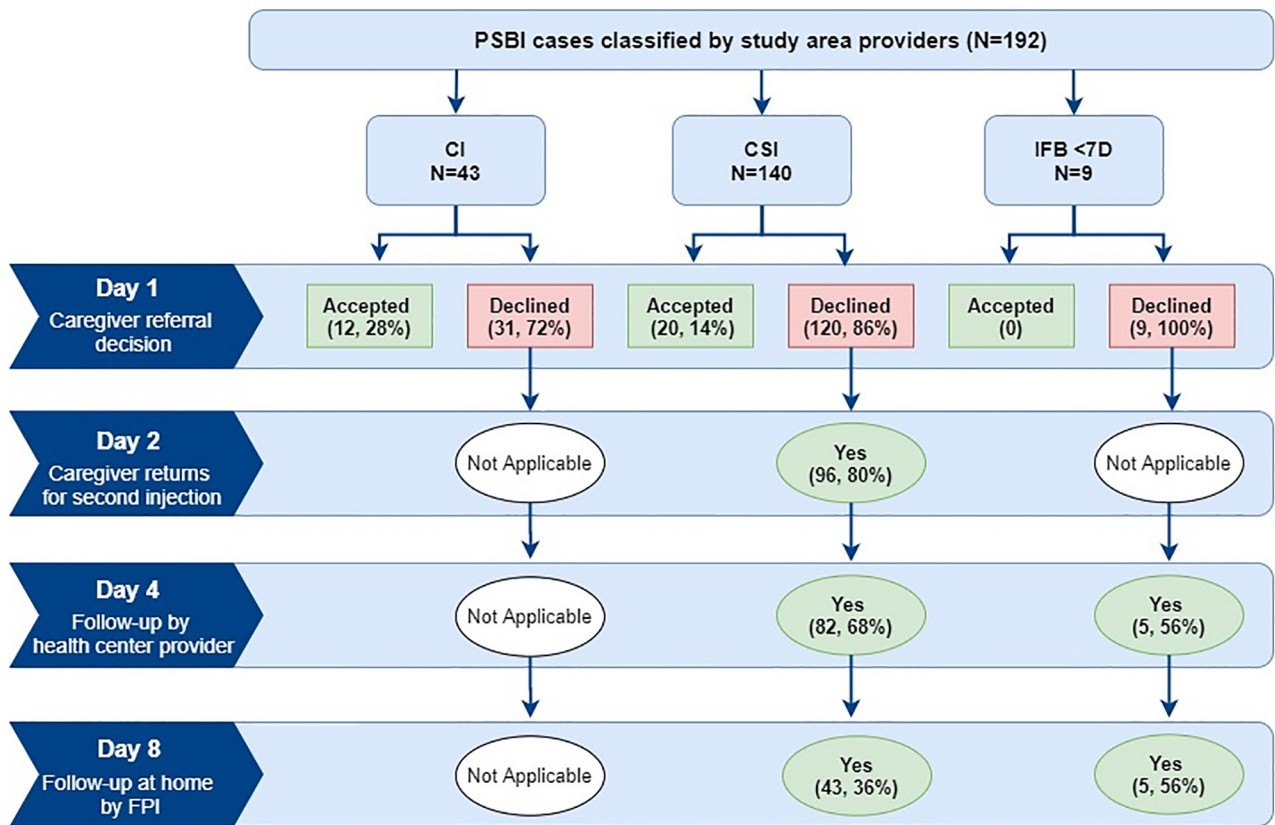

**Fig 3. Management of PSBI cases that require referral and follow-up during the treatment period.**

name, even our husbands. He is very familiar." Mothers in our study all sought care from a study area health center and frequently reported the village doctor as their source of referral due to severity of the illness (e.g., pneumonia, convulsions), or if the illness persisted after the village doctor's treatment. Economic factors (e.g., consultation fee, cost of medicines, travel costs) were key drivers in the family's decision to choose public versus private sector facilities.

Mothers in interviews and group discussions often reported choosing the union-level health centers versus the sub-district hospital when the health center was closer to the community and they trusted the provider—often referring to him or her by name. As one mother said in a group discussion, "We think the treatment [the] doctor provides is trustworthy. That's why we come to UH&FWC [health center]." According to mothers, in group discussions, the quality of care at the government primary health centers can vary by day and location. Caregivers, in interviews and group discussions, mentioned organizational factors (e.g., inconsistent availability of providers, long wait-times, medicine stock-outs) for dissatisfaction with public sector services, but reported these barriers occurred less frequently at the union-level health centers than at the sub-district hospitals.

Providers acknowledged that the guidelines provide an opportunity to improve families' access to and affordability of treatment. As one provider noted in a group discussion,

*In this treatment method, cost is minimal. Earlier, it was very difficult for a poor family to bear the treatment cost.*

–Provider in group discussion

**Table 2. Results of qualitative investigation into reasons for high and low values of caregiver acceptability for key components of the guidelines.**

| Quantitative Results | Qualitative Themes | | Provider Adaption to Strategies |
|---|---|---|---|
| | *Facilitators* | *Barriers* | |
| *Decision to seek care and experience at the public health center* | | | |
| Only 16.3% of the expected number of PSBI cases sought care from the study area health centers | • Caregivers reported seeking care from health centers when they had trust in the provider<br>• Village doctors were often the first source of care and frequently reported as a source of referral to the health center when their treatment failed | • Previous negative experiences with public sector care, including inconsistent availability of providers and lack of medicines, discouraged care-seeking | • Providers encouraged caregivers to publicize services available at health centers |
| *Referral feasibility* | | | |
| Referral was not feasible for 83.3% (N = 160/192) of PSBI cases | • Providers and caregivers reported outpatient treatment with fewer injections and oral antibiotics was more affordable and acceptable than inpatient care | • Some caregivers reported not accepting referral because consent was not obtained from their husband, mother-in-law, or other influential family member<br>• Providers and caregivers reported previous experiences with disrespectful care and inconsistent availability of medicines at the sub-district hospital discouraged future care-seeking from this facility | • Some providers telephoned the sub-district hospital to advise the families were coming and followed-up with caregivers to check on the baby<br>• Caregivers reported fewer delays at the hospital when the provider called ahead and appreciated follow-up from the provider |
| 16.7% (N = 32/192) of PSBI cases families accepted referral | • Caregivers reported they would seek care if they believed the illness to be serious<br>• Many providers reported giving caregivers referral slips and their phone numbers for follow-up | • Referral is not routinely tracked by providers or recorded in the registers, so these infants may be more likely to be lost to follow-up | |
| *Simplified treatment regimen and return for second day injection* | | | |
| 80% of infants with CSI for whom referral was not feasible returned to the health center for the for the second injection on the next day | • Simplified treatment was more affordable and acceptable for continued care than inpatient treatment | • Providers and caregivers reported caregivers did not return due to fear around injections, perceptions that the infant's illness did not warrant a return visit/second injection, or permission was not obtained from an influential family member<br>• Logistical barriers posed a challenge to follow-up when the visit fell on a weekend or a day when the provider was unavailable due to absence or training | • When the provider knew it would not be possible for the caregiver to return the next day, often due to logistical barriers, some providers gave caregivers the second gentamicin injection, to be administered by a village doctor, or instructed them to purchase the injection at a pharmacy |
| *Fourth day follow-up for PSBI cases receiving simplified treatment* | | | |
| 67.4% (N = 87/129) of PSBI cases receiving outpatient treatment received follow-up on the fourth day | • Some providers said they requested the caregivers to return to the facility on day 4 for clinical assessment and would call if the caregiver did not return | • Some providers reported not initiating follow-up, but relying on the caregivers to return to the facility or call if the baby's condition did not improve | • Some providers said they requested the caregivers to return to the facility on day 4 for clinical assessment and would call if the caregiver did not return<br>• Some caregivers reported providing enough oral amoxicillin to the caregiver until day 4, and then requesting they return for the complete regimen (which is not advisable) |

Mothers were aware that services and medicines at government hospitals should be free of cost, but some reported paying a *"visit fee"* ranging from 10–500 Taka (0.12 to 6.00 USD). One mother reported asking the provider about the fee, smiling as she recounted her story,

> *He said, 'We need money to bear the costs' I said, 'Sir, I thought it is government hospital. Why should I give you money?' He said, 'The government bears the cost of medicine only. There are some other costs too.' I said, 'Is it so? Sir, I don't have money with me today. I didn't know. I'll bring next time when I'll come. During the [follow-up] visit, I gave 50 Taka to Sir and said, 'Sir, please have it for tea.'*

> –Mother of sick young infant

Providers did not discuss requesting a visit fee or accepting payment from the caregivers seen outside their private practice. When discussing the caregivers' satisfaction with treatment, one provider in a group discussion said,

> *Being pleased they like to give us some money. But I tell them that this treatment is absolutely free for you. If any babies around you become ill, inform this news to them. Thus, we publicize.*

> —Provider in group discussion

Providers in interviews and group discussions said they regarded the revised guidelines as an opportunity to improve care-seeking from these health centers and encouraged caregivers to share their positive experiences with family members and others in the community.

**Referral feasibility for families of infants with PSBI.**   Our analysis of health center records indicate referral to the sub-district hospital was not feasible for many families (83.3%; N = 160/192). According to providers, in interviews and group discussions, caregivers do not comply with referral primarily due to cost, household responsibilities, needing their husband's consent, and lack of understanding about the severity of the infant's condition. Providers in interviews also acknowledged families' past negative experiences at the sub-district hospital as an important barrier to referral feasibility. One provider noted,

> *There is lack of cordiality to provide service in the higher health care center. As a whole it is seen that they [families] have bitter experience. For these reasons they do not like to go there.*

> —Provider in interview

When probed on referral feasibility, caregivers of sick infants in interviews, acknowledged economic hardships associated with accepting referral, but insisted these factors would not stop them from seeking higher level care for serious illnesses (e.g., pneumonia). For example, one mother said,

> "*He [SACMO] told me to take it [baby] to sub-district hospital, I agreed. I was ready to take any risk for my child. I didn't worry about money. But I wanted my baby to be well again.*"

> –Mother of sick infant

Some caregivers, in interviews, reported not accepting referral or delays in reaching the referral facility because consent of their husband, mother, or mother-in-law was needed. More often caregivers, in interviews and group discussions, cited inconsistent availability of services and community distrust in government doctors as reasons for not accepting referral to the

sub-district hospital. Specifically, mothers reported previous dissatisfaction with government hospital services when their expectations were not met due to doctors not being available during regular service hours, long wait times, high consultation fees, medicine stock-outs, and doctors "misbehaving." When discussing her experience seeking care for her infant at the sub-district hospital, one mother said,

> *Suppose they talk with us angrily, 'There is no medicine. Why have you come here? Government does not give us medicines.' When they tell us these, we get hurt. So we do not go.*

–Mother of infant in group discussion

Caregivers, in group discussions and interviews, often referred to government hospital providers as *"bad doctors."* This label, however, was not assigned based on a perceived lack of technical knowledge or skills. Rather, mothers explained they considered them to be "bad doctors" because of their disrespectful demeanor. Caregivers, in one group discussion, voiced agreement when one mother explained,

> *The doctors at government hospitals have so mean behavior, they feel annoyed when visited, won't speak two words in place of one... But they are not bad as doctors. It is not like they don't have proper medical knowledge. I don't go to them for their behavior.*

—Mother of infant in group discussion

Occasionally mothers accepting referral advice reported choosing to seek care from private hospitals instead of the government hospitals due to the perceived higher quality of care.

When probed on facilitating referral for the 16.7% (N = 32/192) of PSBI cases whose families accepted, providers reported mixed perceptions of their responsibility to follow-up with these families. Providers, in interviews, often reported giving the family a referral slip, their phone number, and the address and phone number of the referral facility. Some providers, however, also reported calling the doctor at the referral facility to say the family was on the way, then later calling the family to ensure they reached the facility. In these cases, the provider reported knowing the outcome of the patient, although there is not a field to record the outcomes of referred cases in the register. Caregivers that accepted referral reported in interviews that they faced fewer delays at the sub-district hospital when the provider called ahead, and felt their infant's health was valued when a provider—either the SACMO or FWV—called to check on the baby's condition.

**Simplified antibiotic regimen and caregiver return for second day injection.** Providers, in interviews and group discussions, highlighted the positives of the simplified antibiotic regimen for caregivers including mothers' preference for fewer injections and treatment with oral antibiotics on an outpatient basis, cost-savings for families, and less time away from their household and other children. When comparing the simplified regimen to standard inpatient treatment, one provider noted,

> *No mother wants her child to be pricked again and again. The revolutionary thing here is one injection for two days.*

–Provider in group discussion

According to providers, mothers do not return to for the second injection due to lack of knowledge and sociocultural factors, which placed onus on the caregiver for "not understanding many things," being "superstitious," and having "fear of modern treatment." Specifically,

providers cited caregiver concerns that the baby's symptoms are not serious enough to warrant the injection and fear that the baby's condition would worsen after receiving the injection. For example, one provider highlighted,

> *We have to make the mothers understand. . .They think that the child might die after taking injection. . . They actually don't know why [a] injection is being provided.*

–Provider in interview

Caregivers, in interviews and group discussions, were cognizant of the value of modern medicine to treat the sick infant especially when "Ayurvedic"—or traditional medicine—had not worked. However, caregivers indicated a clear preference for oral antibiotics given at home versus injections provided at facilities. Some caregivers, in interviews, expressed not returning for the second injection because permission of their husband, mother or mother-in-law was not granted for the return visit and/or injection. Mothers also discussed fears around providing antibiotic injections to young infants and locally accepted beliefs that these medicines could weaken the baby. As one mother described,

> *They gave the injection and I returned home. . .After giving medicines it [baby] became weak. I thought it was the side effects of the injection.*

—Mother of sick young infant

Providers also discussed organizational barriers to the second day injection when the scheduled return visit occurred on a weekend, or when the provider would be absent from the facility due to other duties (e.g., training, supervision meeting at the sub-district hospital). In the provider's absence, the guidelines allow for the FWV to provide the second injection, but providers reported this rarely occurred. Rather, some providers reported dispensing the gentamicin injection to the caregiver and requesting they take it to their village doctor to be "pushed," or writing a prescription to be purchased at an outside pharmacy. Some caregivers of CSI cases confirmed receiving this guidance from the provider and in these cases confirmed visiting a pharmacy, or the village doctor, to have the injection administered. Of the few caregivers that described this experience, some expressed favorable reactions to continuing care with their village doctor closer to their home, while others highlighted potential barriers to this strategy. For example, one mother described the delay she faced when instructed to obtain the second injection from an outside pharmacy,

> *[SACMO] advised me to take the 2nd dose of injection from any pharmacy on the 2nd day. The reason was that [SACMO] would be on training on that day. . .At first we went to pharmacy. They said they cannot give the injection. Then, going to the [Bazar], we got the injection from another pharmacy.*

–Mother of sick infant in interview

When probed on strategies to improve acceptability of the guidelines, providers, in interviews and group discussions, discussed the need for strengthening their counseling and communication skills with caregivers to improve compliance with referral and return visits to the health center, and acknowledged that counseling should include other family members.

Follow-up on the fourth day for infants receiving outpatient simplified antibiotic treatment. Follow-up of PSBI cases on the fourth day of simplified antibiotic regimen is critical to ensure safe and effective treatment [13]. The Bangladesh National Guidelines specify this follow-up

be initiated by phone between the provider and caregiver [25]. Based on our review of facility records, we found about two-thirds (67.4%; N = 87/129) of sick infants received follow-up on the fourth day of treatment. Providers, in group discussions and interviews, agreed with the importance of this visit for monitoring the infant's recovery, but reported various strategies for completing this visit. Some providers reported in interviews that they did not agree with initiating follow-up over the phone, but instead requested the family to return to the facility on the fourth day, or said they would travel to the family's home, so they could clinically assess the child. As one provider described,

> *Calling on 4th day to follow-up, I can't accept this. This is not right following-up over mobile phone. For example, a patient has arrived with low temperature if I don't check his temperature then how can I tell that his condition is improved or not? . . .So I tell them [families] to come directly rather than talking over phone.*
>
> —Provider in interview

When probed on the feasibility of families' return to the health center, providers stated that caregivers often comply. If the caregiver does not agree to return, then providers said they request the caregiver to call or the provider would initiate follow-up by phone. Occasionally, providers in interviews discussed requiring families to return on the fourth day to obtain the full course of oral amoxicillin. One provider discussed using this technique to promote caregiver compliance, "sometimes we state, 'Here is a phial of medicine. Kindly come to see me on the 4th day and collect another phial of medicine too'."

Caregivers of infants receiving outpatient treatment confirmed in interviews that they received varying instructions from the providers regarding follow-up on the fourth day of treatment including phone calls and return visits. Caregivers that participated in this follow-up reported returning to the health center or discussing the infant's condition with the SACMO or FWV over the phone. All caregivers that received follow-up were satisfied with this aspect of care. When probed on the content of the day 4 follow-up phone call, one mother recalled,

> *She asked me how was my baby, whether I have fed my baby medicine and whether the 2nd dose injection was administered. . .I liked the treatment of [SACMO] very much. In a short period, I got very good treatment.*
>
> —Mother in interview

Barriers to follow-up, according to caregivers in interviews, included the provider not initiating the phone call or the caregiver not being given the phone number of the provider. Caregivers said they appreciated the provider giving their mobile phone number and indicated they would call him/her directly when they had questions or to setup an appointment.

## Discussion

This study aimed to explore acceptability of the infection management guidelines from the perception of caregivers and providers during the first year of implementation in primary health facilities in Bangladesh. Few caregivers agreed to referral to the sub-district hospital, suggesting low acceptability of this option for continued care and reiterating the value of the option for simplified antibiotic treatment. Caregivers indicate distrust in hospital doctors, inconsistent availability of medicines, and financial constraints as the primary barriers to referral feasibility. However, caregivers insisted they would seek higher level care, from either the public or private sector, when they believed the infants' illness was severe. Providers and

caregivers indicated high acceptability of simplified antibiotic treatment for infants receiving outpatient treatment, which they attributed to caregiver preference for providing oral antibiotics at home versus continued parenteral treatment at the hospital, reduced medical and travel costs, and less time away from their household and other children. More than three-quarters of infants with clinical severe infection for whom referral was not feasible returned to the facility for the second injection. Providers and caregivers attributed gaps in antibiotic treatment to caregiver concerns around providing injections to young infants and unavailability of the provider. Some providers reported developing local solutions—including engaging village doctors in treatment of the infant—to address organizational barriers and promote treatment compliance. Follow-up of young infants receiving simplified treatment is critical for monitoring the safety of the regimen, but only about two-thirds of families received follow-up from the provider on the fourth day of treatment. Provider deviations from the guidelines (e.g., requesting the caregiver return to the facility or initiate the phone call to the provider) may have contributed to gaps in follow-up since the responsibility of communication was deferred from the provider to the caregivers. Mothers reported greater satisfaction with care when they had a good interpersonal relationship and communication with the provider. These findings suggest strengthening providers' interpersonal skills—including training on counseling that is culturally-sensitive—and reinforcing the responsibility of the provider to initiate and continue follow-up could improve compliance and acceptability of the guidelines.

We estimated that only 16.3% [95% CI: 14.4, 18.5] of the expected PSBI cases sought care from the study area health centers during our study period. Prior studies in this population documented that caregivers' lack of recognition of signs of newborn illness [49] and care-seeking from the private sector contribute to low care-seeking from the public sector [50, 51], but we were not able to examine these factors in our quantitative data. Qualitative data allowed us insight into care-seeking decisions, including caregivers' preference for private providers due to perceived higher quality of care compared to public sector facilities. Informal providers were often indicated as the preferred first choice in care due to their ability to meet the caregivers' emotional and social needs, reliability in providing drugs, and accessibility. Our findings are consistent with other studies that have found village doctors as an important first source of care and referral in the community [18, 27, 29, 52].

Additionally, caregivers and providers discussed utilizing village doctors for administering the second injection when mothers were unable to return to the facility due to unavailability of the provider. In areas suffering from severe health worker shortages, like rural Bangladesh, engaging informal providers in referral and management of sick young infants may feasibly improve reach and acceptability of the guidelines for caregivers [18–20, 25]. However, previous efforts to engage informal providers in health service delivery have encountered complex barriers to implementation including lack of appropriate training, high rates of inappropriate prescribing due to market influences, and the absence of regulation and monitoring systems [18, 27, 36, 53, 54]. For example, one study assessing the feasibility of engaging village doctors to implement Community-based Integrated Management of Childhood Illness (C-IMCI) guidelines in rural Bangladesh found village doctors' knowledge could be improved and retained through training and routine supervision [18]. Despite their increase in knowledge, however, village doctors still engaged in inappropriate prescribing practices for children—especially for antibiotics. Authors suggested inappropriate prescribing behavior was likely influenced by these practitioners' reliance on profits from drug sales and incentives from pharmaceutical companies, highlighting some of the complexities associated with these types of interventions [18]. However, the role of village doctors as the predominant providers for the poor in this context cannot be ignored and our findings suggest they are already integrated in infection management practices for young infants in the community. Previous research has

promoted public-private partnerships (PPP) to improve nationwide coverage of emergency obstetric and newborn care in Bangladesh [55]. Future studies are needed to examine the potential for engaging informal and formal private providers, including the potential for fostering public-private partnerships (PPP) in this context [18, 53, 54]. Previous research has promoted PPP to improve nationwide coverage of emergency obstetric and newborn care in Bangladesh [55]. Additionally, public sector providers' opinions around engaging with private providers as allies should be explored as this has previously been identified as a barrier to implementation of PPP [56].

Our findings suggest referral feasibility is complex for both the families and health system contributing to life-threatening delays in care. Previous research in rural Bangladesh found acceptance of hospital admission for infants with clinical signs of severe infection at the sub-district hospital's outpatient department was less than 20% [57]. Referral acceptance in our study was lower (14%) for infants with clinical signs of severe infection. Families with critically ill infants accepted referral more frequently than infants with less severe signs of PSBI, but acceptance was still low (28%), resulting in 72% (N = 31/43) of critically ill infants not receiving continued treatment or seeking care from the private sector—formal or informal—despite knowing there would be additional out-of-pocket costs. Caregivers' previous experiences with disrespectful treatment by providers and inconsistent availability of medicines at sub-district hospitals negatively affected referral feasibility. A study with sub-district hospital providers in rural Bangladesh found more than one quarter of the respondents did not believe the sub-district hospital was the right place to manage sick newborns, so they preferred to refer the family to a higher-level facility [58]. Strengthening care at public sector referral facilities is needed to promote high quality, timely care for young infants with PSBI requiring inpatient treatment. At the union-level, providers should provide referral slips and follow-up with families to ensure they reached the referral facility, but follow-up is not recorded in the register and cases are not routinely tracked. We found providers reported varying levels of motivation to facilitate referral as demonstrated by connecting with doctors at the sub-district hospital and following up with families to ensure they reached the facility. The few caregivers that described this level of referral facilitation reported improved experiences with care at the hospital and felt their baby's health was valued. As the guidelines are scaled-up, future implementation activities should aim to improve the quality of care at sub-district hospitals, establish systems for tracking referral cases, and reinforce the union-level providers' role to follow-up with referred cases to ensure they reach the higher-level facility.

We found caregivers' perception of the interpersonal nature of care—including communication and trust in providers—were influential in caregivers' acceptability of simplified antibiotic treatment and follow-up. Familial and cultural factors also influenced caregivers' decision to accept referral and return for the second day injection. Our findings are consistent with previous studies on quality of care in Bangladesh echoing the challenges of achieving optimal care that meets both medical and psychosocial needs of the users [18, 27, 36, 59]. Providers have suggested that improved training in counseling of caregivers, husbands, and other influential family members may improve acceptability and compliance with return visits for the second injection. Some providers also discussed developing local solutions to improve treatment compliance and the technical quality of follow-up on the fourth day. It has been suggested by Proctor, that providers' ability to adapt an intervention for local use may increase its acceptability [34, 60]. For example, in the case of fourth day follow-up, providers' requests for families to return to the facility for a clinical visit may improve the acceptability of the guidelines. However, further research into the feasibility of this strategy is needed. Additionally, training and program feedback should reinforce the importance of giving the full course of antibiotics on the day of assessment and responsibility of the provider to initiate follow-up on the fourth day [25].

This mixed methods analysis presents findings on acceptability of the guidelines and identifies barriers and facilitators to simplified treatment and referral feasibility. However, our study had several limitations including lack of direct observations of care. The estimated incidence of PSBI in young infants (95.4/1000) [45] in this setting, coupled with low care-seeking rates from the primary health facilities, led us to expect few infants would seek care at the study area health centers during the initial implementation period. Thus, direct observations of care were not feasible in this study. We were limited to analysis of facility records and qualitative interviews to assess compliance with the guidelines, which did not include caregiver characteristics nor important historical factors (e.g., previous infant death and birth order). Our study did not include implementation support activities or data collection at the sub-district hospital (i.e., UHC)—the recommended referral facility. Thus, we do not have data from the referral facilities to assess referral compliance, quality of care provided, or treatment outcomes for families accepting referral. Given referral complexities identified in this study, we anticipate referral compliance to be lower than acceptance rates recorded in the union-level health center records. Future studies should include data collection at public sector referral facilities to better understand barriers to referral compliance, quality of care, and identify opportunities for strengthening management of newborn infections. The eighth day visit is important for assessing treatment compliance and outcomes of infants receiving simplified antibiotic treatment. However, few infants received the day 8 follow-up visit, which limited our ability to explore the acceptability of this visit in qualitative interviews. Future analyses should include interviews with health workers responsible for this visit (i.e., FPIs) and explore barriers to completion. Finally, we found very few infants aged 0–6 days with IFB sought care at study area health centers, which limited our ability to assess caregiver acceptability of simplified antibiotic treatment for these cases.

There is growing interest around the importance of clients' perceived quality of care in facilities, including the components of respectful care, communication, and responsiveness of providers [61–64]. Similar to other studies, we found caregivers' perceptions of quality of care influences care decisions and may be shaped by community, family and societal expectations and values [61, 64–66]. The health provider and caregiver's interpersonal relationship influenced caregivers' decision to seek care and acceptability of the guidelines—including compliance with referral and simplified antibiotic treatment—which may affect treatment outcomes. Future trainings of providers should discuss strategies for including influential family members in the infant's care, incorporate culturally sensitive counseling messages (e.g., acknowledging caregiver fears around injections), and reinforce the responsibility of the provider to initiate follow-up communication (e.g., referred cases, fourth day phone call) with the caregiver. Local solutions described by providers in our study—including requesting caregivers return for a clinical visit on the fourth day and engaging village doctors in providing the second gentamicin injection—require further examination in this context to assess the safety and potential value of these strategies. Inconsistent tracking of referral cases and perceived poor quality of care from public sector facilities are major barriers to referral acceptability, especially for critically ill infants who are not eligible for the simplified regimen. As the guidelines are scaled-up, future implementation activities should aim to improve the quality of care at sub-district hospitals and strengthen linkages between public sector primary health facilities at the union- and sub-district levels to ensure seriously ill infants receive continued care.

## Supporting information

**S1 Table. Adapted SEM to assess multiple levels of influence on caregiver acceptability of guidelines.**
(PDF)

**S1 File. In-depth interview (IDI) guide: Health providers (SACMO).**
(PDF)

**S2 File. Focus group discussion (FGD) guide: Health providers (SACMO).**
(PDF)

**S3 File. Follow-up question guide: Health providers (SACMO).**
(PDF)

**S4 File. In-depth interview (IDI) guide: Caregivers of sick young infants.**
(PDF)

**S5 File. Focus group discussion (FGD) guide: Caregivers of young infants.**
(PDF)

## Acknowledgments

We acknowledge the contribution of the study participants and the dedication of Projahnmo and MaMoni HSS field teams. Projahnmo is a research partnership of Johns Hopkins University, the Bangladesh Ministry of Health and Family Welfare and other Bangladeshi institutions including ICDDR,B, and Shimantik. We are grateful to USAID for their support to this project, and our technical advisors at WHO and the Johns Hopkins Bloomberg School of Public Health.

## Author Contributions

**Conceptualization:** Dipak K. Mitra, ASM Nawshad Uddin Ahmed, Mohammod Shahidullah, Abdullah H. Baqui.

**Data curation:** Jennifer A. Applegate, Meagan Harrison, Mahfuza Mousumi, Nazma Begum, Mamun Ibne Moin, Taufique Joarder.

**Formal analysis:** Jennifer A. Applegate, Meagan Harrison.

**Funding acquisition:** Abdullah H. Baqui.

**Investigation:** Jennifer A. Applegate, Salahuddin Ahmed, Jennifer Callaghan-Koru, Mahfuza Mousumi, Taufique Joarder, Sabbir Ahmed, Joby George, ASM Nawshad Uddin Ahmed, Abdullah H. Baqui.

**Methodology:** Jennifer A. Applegate, Jennifer Callaghan-Koru, Dipak K. Mitra, Abdullah H. Baqui.

**Project administration:** Salahuddin Ahmed, Mahfuza Mousumi, Nazma Begum, Sabbir Ahmed, Joby George.

**Supervision:** Joby George, Abdullah H. Baqui.

**Writing – original draft:** Jennifer A. Applegate.

**Writing – review & editing:** Meagan Harrison, Jennifer Callaghan-Koru, Taufique Joarder, Abdullah H. Baqui.

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
