## [Decision Letter · Decision Letter 0]

13 Jan 2020

PONE-D-19-30500

Caregiver acceptability of the guidelines for managing young infants with possible serious bacterial infections (PSBI) in primary care facilities in rural Bangladesh: a mixed methods implementation research study

PLOS ONE

Dear Professor Baqui,

Thank you for submitting your manuscript to PLOS ONE. After careful consideration, we feel that it has merit but does not fully meet PLOS ONE’s publication criteria as it currently stands. Therefore, we invite you to submit a revised version of the manuscript that addresses the points raised during the review process.

We would appreciate receiving your revised manuscript by Feb 14 2020 11:59PM. To enhance the reproducibility of your results, we recommend that if applicable you deposit your laboratory protocols in protocols.io, where a protocol can be assigned its own identifier (DOI) such that it can be cited independently in the future. For instructions see: http://journals.plos.org/plosone/s/submission-guidelines#loc-laboratory-protocols

We look forward to receiving your revised manuscript.

Kind regards,

Wen-Jun Tu

Academic Editor

PLOS ONE

Additional Editor Comments:

In order to provide a more complete information to our readers on the topic, we would like to emphasize the importance to cross referencing very recent material on the same topic published in "PLoS ONE ". Therefore, it would be highly appreciated if you would check the contents published in the last two years of "PLoS ONE" (https://journals.plos.org/plosone/) and add all material relevant to your article to the reference list.

Journal Requirements:

2. Please include additional information regarding the interview guide used in the study and ensure that you have provided sufficient details that others could replicate the analyses. For instance, if you developed a guide as part of this study and it is not under a copyright more restrictive than CC-BY, please include a copy, in both the original language and English, as Supporting Information. In addition, please provide details of any pretesting of this guide. Furthermore, please indicate how verbal consent was documented and witnessed.

6. Please include captions for your Supporting Information files at the end of your manuscript, and update any in-text citations to match accordingly. Please see our Supporting Information guidelines for more information: http://journals.plos.org/plosone/s/supporting-information

Reviewers' comments:

Reviewer's Responses to Questions

**Comments to the Author**

1. Is the manuscript technically sound, and do the data support the conclusions?

Reviewer #1: Yes

2. Has the statistical analysis been performed appropriately and rigorously? 

Reviewer #1: Yes

3. Have the authors made all data underlying the findings in their manuscript fully available?

Reviewer #1: Yes

4. Is the manuscript presented in an intelligible fashion and written in standard English?

Reviewer #1: Yes

5. Review Comments to the Author

Reviewer #1: This is a well-written, thoughtful paper and the literature cited considerably enhances our understanding of the findings.

I have a few comments on the analysis and results, and several copy editing points:

1. A descriptive analysis of symptom types and infant demographics is provided but no data is presented on caregiver characteristics (e.g., age of mother) or other potentially significant historical factors (e.g., previous infant death; birth order), whereas you present qualitative information (and literature) implying that caregiver characteristics may be significant determinants of the outcomes of interest. This should be commented on.

2. No secondary analysis is presented looking for associations of any baseline characteristics (predictor variables) and the outcomes of interest. Was this due to small numbers?

3. The relatively low proportion seen (16.3%) of the expected number of infants with PSBI is insufficiently explained. How much of this reflects lack of recognition of symptoms by caregivers? How much reflects care seeking in the private sector versus mortality at home prior to care seeking? This was not addressed in the discussion but would have implications for systematic improvement.

4. Copy editing points (by line number in manuscript):

Line 163: MOHFW is not defined here or later in the manuscript

Line 167: SACMO is not defined "

Line 178: I think you mean 'Isolated fast breathing' (not Isolate fast breathing)

Lines 498/499: this appears to be a mistranscription; phial is a variant of vial - it should not be 'file'

Line 558: should be 'predominant' - the adjectival form, not predominate - the verb form

Line 604: I think you mean "...giving the full course of oral antibiotics on the day of assessment..." (not dose of antibiotics)

6. PLOS authors have the option to publish the peer review history of their article (what does this mean?). If published, this will include your full peer review and any attached files.

Reviewer #1: No

---

## [Author Response · Author response to Decision Letter 0]

29 Feb 2020

Dear Editors,

Thank you for the opportunity to revise our manuscript for reconsideration by PLoS One. We appreciate the comments provided by the Reviewers and their recommendations for strengthening the paper. We have responded to each of the comments and made edits to the manuscript text, tables and figures accordingly. Please see below our itemized responses to the reviewers comments:

1. Please include captions for your Supporting Information files at the end of your manuscript, and update any in-text citations to match accordingly. Please see our Supporting Information guidelines for more information: http://journals.plos.org/plosone/s/supporting-information.

• Response: Thank you, we have included captions for our Supporting Information files at the end of the manuscript and updated the in-text citations to match accordingly. 

2. A descriptive analysis of symptom types and infant demographics is provided but no data is presented on caregiver characteristics (e.g., age of mother) or other potentially significant historical factors (e.g., previous infant death; birth order), whereas you present qualitative information (and literature) implying that caregiver characteristics may be significant determinants of the outcomes of interest. This should be commented on.

• Response: Thank you, for this observation. Data on infant’s demographics and symptom type were available for collection from the young infant registers in the study area health facilities. We agree that caregiver characteristics and birth history are important predictors of newborn infections and access to care. Unfortunately, we do not have data on caregivers’ individual characteristics including birth history. All women were of reproductive age (13-49 years), which we have added to the Results section. We have also acknowledged this as a limitation of the study data. 

3. No secondary analysis is presented looking for associations of any baseline characteristics (predictor variables) and the outcomes of interest. Was this due to small numbers?

• Response: We appreciate this insightful comment. As noted in the above response, we do not have baseline characteristics available for the caregivers. Our quantitative data was extracted from health facility records which only included characteristics for the sick infants. We reviewed our analysis for the potential of examining associations between infant characteristics and our outcomes of interests. As observed by the Reviewer, our quantitative analysis is limited by a small sample size of infants with PSBI. Case management is dependent upon the provider’s classification of infection, which further reduced our sample size for examining associations with the outcomes of interest. Therefore, we focused our quantitative analysis on identifying gaps in caregiver acceptance of guideline components and used our qualitative data to help explain our findings [1]. 

4. The relatively low proportion seen (16.3%) of the expected number of infants with PSBI is insufficiently explained. How much of this reflects lack of recognition of symptoms by caregivers? How much reflects care seeking in the private sector versus mortality at home prior to care seeking? This was not addressed in the discussion but would have implications for systematic improvement.

• Response: Thank you for this valuable insight. We agree that the low proportion of the expected number of cases is an important threat to the program and has implications for systematic improvement. Prior studies in this population documented that caregivers’ lack of recognition of signs of newborn illness [2] and care seeking from the private sector contribute to low care-seeking from public healthcare sector [3, 4], but we were not able to capture these factors in our quantitative data sources. Our focus group discussions allowed us insight into care-seeking decisions, including preference for private providers due to perceived higher quality of care compared to public sector facilities. In our Discussion, we have clearly highlighted this finding as a potential explanation for the low proportion of the expected number of PSBI cases in our study area and compared our findings with other recent studies [3-5]. 

5. Copy editing points (by line number in manuscript):

• Line 163: MOHFW is not defined here or later in the manuscript

• Line 167: SACMO is not defined "

• Line 178: I think you mean 'Isolated fast breathing' (not Isolate fast breathing)

• Lines 498/499: this appears to be a mistranscription; phial is a variant of vial - it should not be 'file'

• Line 558: should be 'predominant' - the adjectival form, not predominate - the verb form

• Line 604: I think you mean "...giving the full course of oral antibiotics on the day of assessment..." (not dose of antibiotics) 

Response: Thank you, we have made all the suggested copy-editing points. 

6. Please include additional information regarding the interview guide used in the study and ensure that you have provided sufficient details that others could replicate the analyses. For instance, if you developed a guide as part of this study and it is not under a copyright more restrictive than CC-BY, please include a copy, in both the original language and English, as Supporting Information. In addition, please provide details of any pretesting of this guide. Furthermore, please indicate how verbal consent was documented and witnessed.

• Response: Both focus group discussions (FGDs) and in-depth interviews (IDIs) were conducted with providers and caregivers to assess their perceptions and acceptability of the guidelines using semi-structured interview guides. The interview guides were piloted by the study team prior to rollout of the guidelines and adapted to improve provider and caregiver comprehension of questions. After each study round, we adapted the questionnaire to explore emergent themes. Verbal consent was chosen for caregivers due to low literacy rates in this population. To ensure the caregiver understood the study procedures, our data collectors read the consent forms aloud and provided the caregiver an opportunity to ask questions. Prior to the initiation of the interviews, all consenting participants either signed or provided their thumbprint as a proof of consent. A witness and member of the research team obtaining verbal consent also signed the consent form. We have included these details on the verbal consent process with caregivers in the revised manuscript. As part of this study we developed our semi-structured interview guides and have included these at supporting information as requested.

7. We note that you have stated that you will provide repository information for your data at acceptance. Should your manuscript be accepted for publication, we will hold it until you provide the relevant accession numbers or DOIs necessary to access your data. If you wish to make changes to your Data Availability statement, please describe these changes in your cover letter and we will update your Data Availability statement to reflect the information you provide.

• Response: Thank you, if accepted, we will provide the relevant accession numbers or DOI required to access our data.

8. Please include captions for your Supporting Information files at the end of your manuscript, and update any in-text citations to match accordingly. Please see our Supporting Information guidelines for more information: http://journals.plos.org/plosone/s/supporting-information

• Response: We have included captions for our Supporting Information files at the end of the manuscript and updated in-text citations to match accordingly. 

References

1. Creswell J, Clark VP. Designing and Conducting Mixed Methods Research. 2nd Edition ed. USA: SAGE Publication Inc.; 2011.

2. Choi Y, El Arifeen S, Mannan I, Rahman S, Bari S, Darmstadt G, et al. Can mothers recognize neonatal illness correctly? Comparison of maternal report and assessment by community health workers in rural Bangladesh. Tropical Medicine & International Health. 2010;15(6):743-53.

3. Applegate JA, Ahmed S, Khan MA, Alam S, Kabir N, Islam M, et al. Early implementation of guidelines for managing young infants with possible serious bacterial infection in Bangladesh. BMJ Global Health. 2019;4(6).

4. Baqui AH, McCollum ED, Mahmud A, Roy A, Chowdhury NH, Rafiqullah I, et al. Population-based incidence and serotype distribution of invasive pneumococcal disease prior to introduction of conjugate pneumococcal vaccine in Bangladesh. PloS one. 2020;15(2):e0228799.

5. Guo S, Carvajal-Aguirre L, Victora CG, Barros AJ, Wehrmeister FC, Vidaletti LP, et al. Equitable coverage? The roles of the private and public sectors in providing maternal, newborn and child health interventions in South Asia. BMJ global health. 2019;4(4):e001495.

---

## [Decision Letter · Decision Letter 1]

25 Mar 2020

Caregiver acceptability of the guidelines for managing young infants with possible serious bacterial infections (PSBI) in primary care facilities in rural Bangladesh

PONE-D-19-30500R1

Dear Dr. Baqui,

We are pleased to inform you that your manuscript has been judged scientifically suitable for publication and will be formally accepted for publication once it complies with all outstanding technical requirements.

With kind regards,

Wen-Jun Tu

Academic Editor

PLOS ONE

Additional Editor Comments (optional):

Reviewers' comments:

Reviewer's Responses to Questions

**Comments to the Author**

1. If the authors have adequately addressed your comments raised in a previous round of review and you feel that this manuscript is now acceptable for publication, you may indicate that here to bypass the “Comments to the Author” section, enter your conflict of interest statement in the “Confidential to Editor” section, and submit your "Accept" recommendation.

Reviewer #1: All comments have been addressed

2. Is the manuscript technically sound, and do the data support the conclusions?

Reviewer #1: Yes

3. Has the statistical analysis been performed appropriately and rigorously? 

Reviewer #1: Yes

4. Have the authors made all data underlying the findings in their manuscript fully available?

Reviewer #1: Yes

5. Is the manuscript presented in an intelligible fashion and written in standard English?

Reviewer #1: Yes

6. Review Comments to the Author

Reviewer #1: Thank you for your response to my initial review. This paper is entirely sound and requires only some very minor copy editing as follows:

line 566: sells - should be sales

line 581: ...signs of clinical severe infection.. - should be ..clinical signs of severe infection

lines 610-611: providers’ request for families to return to the facility for a clinical visit may improve their acceptability of the guidelines - should be: ..providers' requests for families to return to the facility for a clinical visit may improve the acceptability of the guidelines.

lines 620-622: this sentence is not grammatically correct nor entirely clear - please revise it.

line 632: is functionality the right word? - Do you mean utility, or value or effectiveness?

line 636: client's should be clients'

7. PLOS authors have the option to publish the peer review history of their article (what does this mean?). If published, this will include your full peer review and any attached files.

Reviewer #1: Yes: Richard A Bedell

---

## [Editor Report · Acceptance letter]

31 Mar 2020

PONE-D-19-30500R1 

Caregiver acceptability of the guidelines for managing young infants with possible serious bacterial infections (PSBI) in primary care facilities in rural Bangladesh 

Dear Dr. Baqui:

I am pleased to inform you that your manuscript has been deemed suitable for publication in PLOS ONE. Congratulations! Your manuscript is now with our production department. 

With kind regards,

on behalf of

Dr. Wen-Jun Tu 

Academic Editor

PLOS ONE